# To Repeat or Not to Repeat: Repetitive Sequences Regulate Genome Stability in *Candida albicans*

**DOI:** 10.3390/genes10110866

**Published:** 2019-10-30

**Authors:** Matthew J. Dunn, Matthew Z. Anderson

**Affiliations:** 1Department of Microbiology, The Ohio State University, Columbus, OH 43210, USA; dunn.658@osu.edu; 2Department of Microbial Infection and Immunity, The Ohio State University, Columbus, OH 43210, USA

**Keywords:** genome stability, telomere, subtelomere, gene family expansion, LTR, MRS, *Candida albicans*

## Abstract

Genome instability often leads to cell death but can also give rise to innovative genotypic and phenotypic variation through mutation and structural rearrangements. Repetitive sequences and chromatin architecture in particular are critical modulators of recombination and mutability. In *Candida albicans*, four major classes of repeats exist in the genome: telomeres, subtelomeres, the major repeat sequence (MRS), and the ribosomal DNA (rDNA) locus. Characterization of these loci has revealed how their structure contributes to recombination and either promotes or restricts sequence evolution. The mechanisms of recombination that give rise to genome instability are known for some of these regions, whereas others are generally unexplored. More recent work has revealed additional repetitive elements, including expanded gene families and centromeric repeats that facilitate recombination and genetic innovation. Together, the repeats facilitate *C. albicans* evolution through construction of novel genotypes that underlie *C. albicans* adaptive potential and promote persistence across its human host.

## 1. Introduction

Considerable variability in genome organization exists across the tree of life, ranging from single circular chromosomes in bacterial species to thousands of linear chromosomes in some ferns [1]. These changes in chromosome complement arise through underlying processes of genome instability, giving way to structural reorganization and locus shuffling. Emergence of novel genetic and karyotypic arrangements can increase phenotypic diversity within a population and provide adaptive solutions to emerging selective pressures. Yet, extreme forms of genome plasticity can lead to DNA fragmentation, chromosome loss, and cell death. Thus, a balance between prohibitive and unrestricted genome instability is required to maintain DNA integrity while allowing evolution to produce novel and potentially advantageous genotypes. 

Repetitive regions of the genome often serve as hotspots of genome rearrangements and evolutionary innovation. These elements can exist as single or multi-copy loci encoding tandemly-repeated identical or near-identical sequences of variable length ranging from one to thousands of nucleotides. By their very nature, repetitive loci are prone to recombination, producing insertion/deletions (indels) and translocations between both adjacent and distal repeats. Recombination within repeats alters copy number variants in addition to promoting de novo mutations due to errors in DNA exchange and repair [2]. In contrast, recombination between distal, non-allelic elements can result in large-scale chromosomal aberrations such as fusion events, truncations, and translocations. Genetic exchange between repeats does not require perfect sequence identity and commonly occurs between imperfect repetitive sequences [3]. Repetitive loci are also particularly prone to replication fork collapse and environmental damaging agents, including oxidative stress and ultraviolet irradiation, that promote further mutation and recombination [4,5,6]. Condensed heterochromatin across repetitive regions of the genome is thought to help shield these regions from DNA damage, but targeted investigation of specific loci has revealed that open chromatin often borders these repetitive loci, suggesting they are susceptible to these same mutagenic forces [7]. Recent work has further complicated the relationship between chromatin architecture and DNA damage susceptibility as the damaging agent and chromatin status of the DNA both contribute to altered susceptibility to double strand breaks, mutation, and recombination [8,9]. Thus, a more complex interplay between the locus and genetic insult determine the likelihood for mutations to accrue and recombination to take place.

Fungi are well established model systems for the investigation of cellular, molecular, and genetic processes [10]. Even prior to the emergence of genome sequencing, fungi were pilot organisms for the analysis of repetitive DNA, including the investigation of cornerstone sequences such as centromeres and telomeres [11,12,13,14,15]. Progression into the genomics era led to rapid advances in the catalogue of available fungal genomes for mining and deciphering common features of functional repetitive sequences [16,17,18]. In particular, budding yeast genomes provided much of the initial understanding of eukaryotic genome architecture and intra-species sequence diversity because of their relative smaller sizes and well-defined gene features. The prevalent fungal commensal and opportunistic human pathogen *Candida albicans* possesses many of these same hallmark attributes. The diploid *C. albicans* genome is comprised of eight chromosomes numbered 1 through 7 by size with the exception of chromosome R (ChrR), which encodes the rDNA locus [19,20,21]. *C. albicans* isolate SC5314, the genome reference strain, possesses approximately 6400 genes over ~14 megabases with one single nucleotide polymorphism (SNP) approximately every 300 basepairs (bp) between chromosome homologs [22,23]. Although *C. albicans* is most commonly isolated as a heterozygous diploid, strains are capable of propagating with altered ploidy, copy number variants, and patterns of heterozygosity [24,25,26]. Some of these genotypic changes confer selective advantages to specific environmental conditions in vitro and in vivo, including the presence of antifungal drugs and the oral cavity [25,27,28,29,30,31,32]. Thus, genomic rearrangements may contribute to the success of *C. albicans* colonization, persistence, and adaptability across host niches.

Four major repetitive regions are present in the *C. albicans* genome: telomeres, subtelomeres, the Major Repeat Sequence (MRS), and the rDNA locus (Figure 1). Additional repetitive sequences are interspersed throughout the genome where they contribute to the collective process of chromosome instability and genome evolution (summarized in Table 1). Here, we describe these repetitive sequences and highlight the role they play in genome dynamics in this important model organism and human fungal pathogen.

## 2. Telomeres and Recombination

*C. albicans* maintains the ends of its linear chromosomes through the use of telomeres that are composed of repeating subunits produced by the enzyme telomerase. The telomeric repeats of most eukaryotes are built from a repeating 6 bp GT motif, but many fungal species including the closely-related Saccharomycotina yeasts use repeats of diverse lengths and complexity [33]. In *C. albicans*, the telomeric repeat is 23 bp (5’-ACTTCTTGGTGTACGGATGTCTA-3’), and has diverged substantially from other *Candida* species by increasing in both length and complexity [34]. Other *Candida* CUG paraphyletic species, such as *Candida guillermondii* and *Debaryomyces hansenii,* decode the “CUG” codon as serine instead of leucine and use shorter telomeric repeats of 5’-ACTGGTGT-3’ and 5’-ATGTTGAGGTGTAGGG-3’, respectively [35,36]. Telomerase reverse transcriptase (TERT) and the telomerase RNA component (TERC) work together to extend telomeres by adding single 23 bp repeats to nascent chromosome ends to form the full telomere, which can range in size from 500 bp to 5 kilobases (kb) in *C. albicans* laboratory isolates [37,38].

### 2.1. Telomere Structure and Maintenance

The telomerase complex, which includes TERT and TERC, forms the basic functional unit required for telomere repeat production and maintenance [39,40,41]. Both TERC and TERT, encoded by *TER1* and *EST2*, respectively, perform repeated rounds of reverse transcription to build G overhangs as part of the end-protective T-loops that prevent telomere attrition [42]. More specifically, Est2 uses the *TER1* RNA as a template to add telomeric repeat subunits to the chromosome end in order to maintain replicative capacity and avoid cell senescence. The dual roles of end protection and telomere repeat addition by Est2 are separable. A catalytically inactive Est2 retains the ability to suppress excessive G-strand accumulation despite being unable to add telomeric repeats [42]. Two additional telomerase protein subunits, Est1 and Est3, also contribute to appropriate telomere structural maintenance by suppressing excessive recombination and preventing telomere loss although their precise molecular mechanisms remain obscure [43].

Maintenance of telomeric repeats in *C. albicans* requires the heterotrimeric CST complex comprised of Cdc13, Stn1, and Ten1. Loss of any of these factors results in dysregulation of telomere length, resulting in either overextended or shrunken telomeres [37]. This contrasts with inactivation of the telomerase complex that only leads to iterative telomere shortening due to the inability to add telomeric repeats. The canonical Ku70/Ku80 complex also controls telomere length and loss of either gene produces heterogeneous telomere lengths indicative of dysfunctional telomerase activity [44,45]. Rap1 (repressor/activator protein 1), an essential protein in telomere maintenance in other budding yeasts, is surprisingly not essential in *C. albicans* despite also functioning in telomere length regulation [46,47]. 

### 2.2. Telomere Replication and Recombination

Telomere maintenance through recombination serves a critical function in chromosome stability but can also contribute directly to genome instability during telomere length shortening. As such, the actions of telomere length control and protection from overhang accumulation must be tightly controlled. Telomere length can be regulated through addition of telomeric repeats following their loss during DNA replication or via recombination within single telomeres or between chromosomes [48]. Alternative lengthening of telomeres (ALT), a process that is telomerase-independent but recombination-dependent, adds telomeric repeats to chromosome ends by using pre-formed telomeres as a template [49]. The 3’ overhang of the extending telomere invades other telomeric DNA that can be either: within the same telomere through telomere looping, another chromosome’s telomere, or an extrachromosomal telomere circle. Telomere circles (t-circles), autonomous plasmids composed entirely of telomere repeat subunits, can be produced as a byproduct of telomere length regulation via intra-chromosomal recombination events [50]. In *C. albicans*, t-circle frequency increased following loss of Ku70, suggesting these proteins suppress intra-telomeric recombination [44]. However, no direct studies of ALT have been performed in *C. albicans* despite the presence of homologous genes that appear capable of conducting these functions (e.g., Rad52 [45,49,51]).

Conserved telomeric proteins are required for telomere length regulation. For example, Rap1 likely plays a role in this process, as its deletion produced aberrant telomere repeat-containing DNA structures and t-circles [47]. Yet, all observed nuclear telomere lengthening events in *C. albicans* have occurred through concatenation of linear elements by *TER1* [52], leaving the function of Rap1 in t-circle production and ALT unclear. Our current understanding of the recombination machinery (e.g., Rad52, etc.,) and precise mechanisms for telomere maintenance are severely deficient in *C. albicans* as little direct experimental work has been performed in recent decades. Investigations into *C. albicans* telomere biology, in particular, stands out for its potential scientific gains given the paucity of existing data combined with the common involvement of telomeres in contributing to genome plasticity. Newly developed linear plasmids containing in *C. albicans* share similar mechanisms of linear telomere formation and maintenance for their retention and propagation and may present as a simplified model to study telomere dynamics in *C. albicans* [53].

## 3. *Candida albicans* Subtelomeres Are Hotspots of Genomic Rearrangements

Subtelomeres are defined as the genomic regions adjacent to telomeric repeats that are enriched for repetitive genetic elements [54]. Repetitive sequences within the subtelomeres include intact transposable elements and expanded gene family members as well as remnants of these functional units following gene disruption or inactivation. Formation of heterochromatin conducive to epigenetic silencing is also a common feature of subtelomeres, although it may be interspersed with euchromatic regions surrounding intact genes. In *C. albicans*, the histone deacetylase Sir2 maintains the heterochromatin within subtelomeres [55,56]. Surprisingly, Sir2 appears to perform these roles without other Sir complex proteins typically required for subtelomeric silencing through “telomere position effect”, as they are absent from the *C. albicans* genome [57]. The *SIR2* paralog, *HST1*, also regulates subtelomeric gene expression, although these effects are less uniform across *C. albicans* subtelomeres with only some loci being affected in a Δ/Δ*hst1* background [58]. These combined effects of heterochromatin formation and gene silencing in *C. albicans* were recently shown to be strongest within the proximal 10–15 kb of the telomere repeats [59]. As such, we choose to define the *C. albicans* subtelomeric region as the most telomere-proximal 15 kb of each chromosome arm.

### 3.1. Subtelomeric Gene Functions

*C. albicans* subtelomeres are relatively gene poor compared to chromosome-internal regions of the genome. Approximately 50 protein-coding genes are annotated within *C. albicans* subtelomeres in the genome reference strain SC5314 based on the above definition from the *Candida* Genome Database [60], but most genes remain largely uncharacterized (Table 2). Most of the subtelomeric genes with functional assignments are either associated with biofilm formation (23 of 55), growth in Spider or other hyphae-inducing media (22 of 55), or have predicted roles in metabolism (21 of 55; Table 2). Coincidently, many of these genes are also assigned roles in virulence either by directly promoting pathogenicity or through adaptive responses promoting persistence across host niches. Genes that may be used in niche adaptation include multiple transporters, cell wall proteins, and metabolite utilization genes.

Approximately one quarter of *C. albicans* subtelomeric genes (13 of 55) belong to the telomere-associated (*TLO*) gene family (Figure 2). *TLO* genes underwent a recent lineage-specific expansion from a single gene in most *Candida* species to 14 paralogs in *C. albicans* and are the only gene family with a significant presence within the subtelomeres [61,62,63]. Each *TLO* contains a MED2 domain, indicating their function as homologs of the Med2 subunit in Mediator, the major eukaryotic transcriptional regulatory complex [63]. *TLO*s participate in regulating a variety of virulence traits, including growth, resistance to stressors, and biofilm formation [64,65]. Strikingly, individual *TLO* genes regulate distinct virulence properties despite sharing >98% nucleotide identity between individual paralogs. 

### 3.2. Repetitive DNA Elements in the Subtelomere

Repetitive elements commonly cluster in subtelomeric regions where they can buffer gene-rich chromosomal interiors from the detrimental effects of telomere length variation [67]. In particular, retrotransposons are often enriched within subtelomeres as new insertions are expected to incur less of a fitness cost in these gene-poor regions and can provide selective advantages by rescuing telomerase-deficient cells [68,69]. Over 350 unique retrotransposon insertions by 34 distinct families of transposable elements have been identified in the *C. albicans* SC5314 genome [70], which can be distinguished by the unique sequences of flanking long terminal repeats (LTRs). Of these 34 families, seven LTR families are found within the *C. albicans* subtelomeres, including an intact copy of Zorro2, the only complete non-LTR retrotransposon in the subtelomeres (Table 3) [70]. Thirteen individual LTRs are annotated in the subtelomeres, ~2.5x higher than the genome average. Importantly, LTR sequences incorporated as part of functional genes (e.g., *TLO*s) are not included among annotated repetitive sequences, suggesting these frequencies are likely an underestimate. Most evidence for transposon integration events in the subtelomeres comes from abandoned LTR repeats that mark previous insertions which subsequently reactivated the intervening transposase to excise itself and reintegrate elsewhere in the genome. In addition to inducing genotypic variation via continual transposon movement, highly abundant non-LTR retrotransposons in *C. albicans* can generate genetic diversity through recombination between dispersed LTR sequences [71].

Additional sequence elements, detected through molecular investigations, exist in *C. albicans* subtelomeres. These elements, the Bermuda Triangle sequence (BTS) and the *TLO* recombination element (TRE), lie immediately centromeric to *TLO* genes (Figure 2). The BTS is defined by a 50 bp sequences which share ~88% sequence similarity across the 11 BTS-containing subtelomeres [72]. Each BTS is encompassed within the longer *TLO* recombination element (TRE). The TRE stretches ~300 bp, beginning immediately following the *TLO* coding sequence and extending towards the centromere [59]. While no clear phylogenetic hierarchy is observed among the BTS sequences, TREs can be organized into three groups based on near complete sequence identity within any single cluster. Both the BTS and TRE overlap with a member of the *tui*-class family of LTR sequences which are annotated in some subtelomeres but missing in others (Figure 2). Given the high sequence homology across the BTS and TRE, it is likely this LTR element is present next to all *TLO* genes with the exception of ChrRR and Chr7R, which both lack the BTS and TRE [56]. 

### 3.3. Subtelomeres Are Prone to Mutation

*C. albicans* subtelomeres experience high rates of recombination, including frequent loss of heterozygosity (LOH). LOH events result from non-reciprocal crossing over or chromosome loss and reduplication of the lost region from the remaining chromosome homolog. Rates of LOH increase along all chromosomes towards the telomeres of *C. albicans* isolates, indicating a general relationship between the distance from the centromere and allelic homozygosis mediated by recombination [73] (Figure 3). Yet, subtelomeric LOH rates increase an order of magnitude compared to centromere-proximal regions interior to the TRE and loss of the TRE in subtelomeres results in a significant decrease in LOH [56]. Integration of the TRE into an exogenous locus increased LOH rates of an adjacent selectable marker [56]. Thus, the TRE sequence is crucial to the elevated recombination rates in *C. albicans* subtelomeres where it suppresses mitotic recombination additively with Sir2 silencing. However, some stressors, including fluconazole, promote such a strong genome instability phenotype that the protective effects of Sir2 can be masked in their presence [56].

Phosphorylation of serine 129 on histone H2A, commonly known as γH2A, is a common marker for heterochromatin in *S. cerevisiae* [74]. Consistent with localizing to heterochromatic regions, γH2A is enriched at subtelomeres, but is also found at origins of replication and convergent genes where it labels double-strand DNA breaks. More specifically, γH2A abundance is statistically enriched on 13 of the 16 *C. albicans* subtelomeres (all except ChrRR, 1R, and 7R). Its conspicuous absence in select subtelomeres is thought to be due to incomplete assembly in the current genome assembly and not the absence of γH2A [75]. These γ-sites mark putative recombination-prone genomic loci that are intrinsically more fragile and susceptible to DNA damage [75].

Subtelomeric recombination directly affects their genic repertoire. Continuous passaging of SC5314-derived strains for 30 months led to the gain and loss of specific *TLO* genes and all telomere-proximal sequences through non-reciprocal recombination events among different chromosome arms [72]. These non-reciprocal exchanges occurred once every 5000 generations on average, with some chromosome arms being favored as either DNA donors or recipients. Bias in amplification and loss of specific subtelomeric sequences due to non-reciprocal DNA exchange suggests that selection favored expansion or loss of certain genomic regions during passaging. Approximately half of the detected recombination events occurred within the BTS element with two additional events initiating in the TRE. Two recombination events took place within *TLO* genes and produced novel *TLO* sequences, highlighting the potential for significant sequence diversity to arise via subtelomeric recombination that can contribute to phenotypic plasticity [72]. 

Genes within the subtelomeres are subject to not only increased rates of recombination, but also increased formation of indels [73]. *TLO*s can be separated into three clades (α, β, and γ) based on sequence diversity that primarily clusters in the 3’ end of the coding sequence. The *TLO*β-clade is unique in containing two major indel events in this 3’ region that distinguish it from the other *TLO* clades. The 3’ end of TLOγ-clade genes uniformly contains a *rho* LTR sequence that likely disrupted an ancestral *TLO*α-clade homolog which then expanded across the subtelomeres through non-reciprocal exchange of this newly-formed TLOγ-clade gene [62]. Further demonstrating the evolutionary potential of continual retrotransposon activity, a subsequent *psi* LTR-containing retrotransposon insertion disrupted *TLO*γ*4* in the SC5314 background, producing a truncated *TLO*γ*4* gene, which is actively transcribed, and a *TLO*ψ*4* pseudogene with no detectable transcript [62,72].

## 4. The *C. albicans* Major Repeat Sequence

Seven of the eight *C. albicans* chromosomes, the exception being Chr3, contain large tracts of repetitive DNA known as the MRS. These regions are comprised of large, variable arrays of the 2 kb repetitive sequence (RPS) unit that are often flanked by the non-repetitive RB2 (6 kb) and HOK (8 kb) elements [21,76]. RPS-1, the main unit of the MRS, is itself comprised of smaller repeat segments of about 80–170 bp, which are then assembled into two families of ordered segments, REP1 and REP3, which both contain the 29 bp COM29 common sequence (5’-CAAAAAAGGCCGTTTTGGCCATAGTTAAG-3’) [77]. Altogether, *C. albicans* possesses nine complete MRS loci, 14 RB2 elements, and two HOK elements (Figure 4). MRS loci contain rare 8-bp SfiI restriction sites that have been used extensively to separate chromosome arms for mapping genetic loci prior to construction of the genome reference sequence and for detection of chromosomal rearrangements by contour clamped homogenous electric field (CHEF) electrophoresis [78].

### Impacts of the MRS on Chromosome Loss and Recombination

Despite the highly nested repeat structure of the MRS sites, recombination frequencies in the MRS are equivalent to the genome average rates of recombination. Thus, the MRS is unlikely to function as a recombination hotspot, but instead often contributes to translocations between heterologous chromosomes [79]. Insertion of a *URA3* selectable marker into the RB2 element of various MRS sites allowed translocation events to be tracked that involved multiple different chromosomes as well as truncations of Chr7 in a few instances [80]. Broad karyotypic surveys across clinical isolates have repeatedly identified chromosomal translocations involving the MRS of different chromosomes, suggesting that these chromosomal rearrangements are rare but stably maintained over longer evolutionary time frames by *C. albicans* isolates [81,82,83]. In particular, the WO-1 strain, in which the white-opaque cell state switch was first described, contains three chromosomal translocations that are coincident with the MRS of different chromosomes [84,85]. 

Intrachromosomal recombination also occurs within the MRS. The XhoI restriction enzyme cuts chromosomal DNA on either side of the MRS that can be used to define MRS size by Southern blotting with a RPS-specific probe [79]. Changes in MRS size can also be observed by CHEF electrophoresis of chromosome homologs that resolve from each other due to repeat expansion or contraction. Detection of these events is more readily observable for smaller chromosomes (e.g., Chr7) that have greater length resolution [83]. Loss of the MRS through targeted deletion suggests that these repeats are not required for gross viability in *C. albicans* [86]. Yet, chromosome loss due to nondisjunction was directly related to MRS size based on selection for loss of Chr5 by growth in sorbose [86]. Chr5 homologs encoding the smaller MRS were preferentially retained compared to homologs with longer MRS regions [79,86]. 

## 5. The rDNA locus

*C. albicans* rDNA repeats are encoded as an array on the right arm of ChrR at the *RDN1* locus. The rDNA locus encompasses the 18s, 5.8s, 25s, and 5s rRNAs that are organized as tandem repeating units. This locus ranges in size from 11.6 kb to 12.5 kb and shifts between 21 and 176 copies of repeating rDNA units across strains and growth conditions [87]. Consequently, the full *RDN1* locus can vary between 244 and 2200 kb or approximately 10% of the total size of the *C. albicans* genome. The presence of the rDNA locus and such massive shifts in chromosome size underlie the distinct name of the encoding chromosome, ChrR [88]. Naming of this chromosome breaks from the numbering system based on length used for all other nuclear chromosomes, as changes in rDNA copy number can greatly alter its size. 

### Recombination between rDNA Repeats

Unequal intrachromosomal recombination events within the rDNA repeats are frequent and produce the large-scale shifts in *RDN1* size. Changes in rDNA repeats contribute to approximately 92% of *C. albicans* chromosomal variation and instability in colony morphology mutants [87]. The frequency of these recombination events lacks precise quantification but can be observed within five cell divisions by next-generation sequencing (unpublished data). Extrachromosomal rDNA circles and linear rDNA plasmids can also be released during recombination [89,90]. The monopolin complex, shown to be important for regulating rDNA and telomeric repeats in *S. cerevisiae* [91], also contributes to *C. albicans* rDNA maintenance. Strains lacking monopolin encoded shorter and more variably-sized rDNA, implicating monopolin in specifically maintaining rDNA length [92].

## 6. Additional Repetitive Sequences Shape *Candida albicans* Genomic Stability

Although telomeres, subtelomeres, the MRS, and the rDNA locus are all clearly defined genomic regions in *C. albicans* that play important roles in genome stability, multiple additional repetitive sequences throughout the genome promote recombination and shape genome evolution.

### 6.1. The Agglutinin-Like Sequence (ALS) Family of Adhesins

Critical to *C. albicans’* success within the human host is its ability to adhere to a wide range of substrates. Colonization and adherence are mediated, in part, by the *ALS* gene family of adhesins. *ALS* genes family members are encoded by eight distinct loci that each produce a glycoprotein capable of forming amyloid-like aggregates [93,94]. Within each *ALS* locus, a conserved Ser/Thr-rich domain provides a platform for intergenic recombination between *ALS* homologs and production of new genetic variants [93,94,95,96]. Indeed, *ALS51* is a product of recombination between the *ALS1* and the *ALS5* loci that is present in multiple clinical isolates of *C. albicans*. Vast allelic differences can also arise through intragenic recombination. Hypermutability of the *ALS7* ORF has produced 60 *ALS7* alleles identified across a collection of 66 strains [97]. Variant alleles arise via frequent recombination within tandem repeats and conserved 3’ repetitive domains. Mutation and LOH rates within this gene family are also remarkably high, comparable to that within the MRS, subtelomeric regions, and the *TLO* gene family [24]. Thus, recombination within and among these cell surface antigens allows for rapid gene turnover and adaptation via cellular adherence to environmental substrates that can promote colonization and persistence across body sites [98]. 

### 6.2. Genome Evolution through Centromere-Associated Repeats

Centromeres in *C. albicans* are unique, gene devoid stretches of DNA typically flanked by inverted repeats [60,99]. Recombination between these flanking repeats can lead to the formation of isochromosomes, which are chromosomes comprised of two identical chromosome arms joined by a functional centromere. Formation of an isochromosome of Chr5L (i5L) commonly follows *C. albicans* exposure to azole-class antifungal drugs [29]. Amplification of the left arm of Chr5 confers azole resistance through the acquisition of extra copies of *ERG11* and *TAC1,* encoding the canonical azole drug target and the transcriptional activator of efflux pumps, respectively [100]. Recombination between the centromeric repeats flanking either side of the Chr5 centromere is responsible for producing i5L [29]. Similarly, clinical isolate P78042, which was isolated with a Chr4 trisomy [25], produced an isochromosome of the right arm of Chr4 during passage in the presence of FLC for 100 generations [101]. These examples highlight the importance of recombination and isochromosome formation through centromere-associated repeats for adaptation of *C. albicans* to antifungal drugs [101]. 

### 6.3. Recombination Facilitated by Cryptic Long Repeats

A recent scan of the *C. albicans* SC5314 genome identified 1974 long repeat sequences, excluding the rDNA, the MRS, subtelomeric repeats, and previously characterized complex tandem repeats [101]. Called repeats were required to have sequence matches of 20 bp or longer, which occur more than once in the genome and accounted for 2.87% of the haploid reference genome altogether. Repeat location did not correlate with GC content or ORF density and many repeats encompassed complete and actively transcribed genes [101]. Recombination at these long repeats produced strains harboring copy number variations, LOH, and chromosomal inversions. Consequently, repetitive sequences scattered throughout the genome must be included as additional arbiters of genome instability [101].

## 7. Conclusions

Thus, repetitive sequences in the *C. albicans* genome promote genome instability through various mechanisms that can range from minor changes in sequence length to massive chromosomal rearrangements. Previous work has focused on defining the characteristics of four major repetitive regions (telomeres, subtelomeres, the MRS, and the rDNA locus), but their contributions to genetic and phenotypic diversity still remain generally unresolved. Furthermore, the importance of less well-understood repetitive sequences to genome evolution and karyotypic innovation are just beginning to emerge. What is clear is that repetitive elements within the *C. albicans* genome produce novel genotypes through changes in DNA sequences or karyotypes that often prove to be beneficial. Some of these changes, such as isochromosome formation during antifungal drug exposure and changes to gene family copy number, are more intuitive, whereas the function of others involving changes in MRS repeat length and rDNA copy number are less easy to grasp. It could be that repeat length modulates the probability of recombination occurring or that the presence of repeats is itself sufficient to promote genome instability. Association of additional repetitive sequences at major regions within the genome (e.g., telomeres, the MRS) suggest that these sites are likely under relaxed selection for new insertions that increase the likelihood of additional recombination events. In contrast, repeats scattered throughout the genome beyond these defined regions may function as back-up sites for genome instability. These cryptic repetitive sequences throughout the *C. albicans* genome may not only promote recombination but reduce the associated fitness costs of novel genotypes by increasing the resolution of affected sequences to only the locus (loci) that is under selection. Regardless, these repetitive sequences collectively construct a dynamic environment of common and infrequent sites of genomic instability to generate genotypic and phenotypic variants that allow *C. albicans* adaptation to its diverse niches across the human host and ultimately promote its success as a commensal and pathogen. 

## Figures and Tables

**Figure 1 genes-10-00866-f001:**
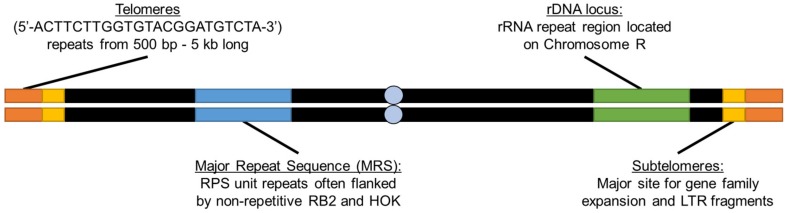
Major classes of genetic repeats in *Candida albicans*. *C. albicans* contains four major categories of repeat sequences: the telomeres that contain multiple copies of a 23 bp repeat; the Major Repeat Sequence (MRS) composed of repetitive sequence (RPS) repeats; the rDNA locus, which encodes the polycistronic rRNA transcripts; and the subtelomeres, a telomere-proximal region containing transposable elements and a gene family expansion. Centromeres are indicated by grey circles.

**Figure 2 genes-10-00866-f002:**
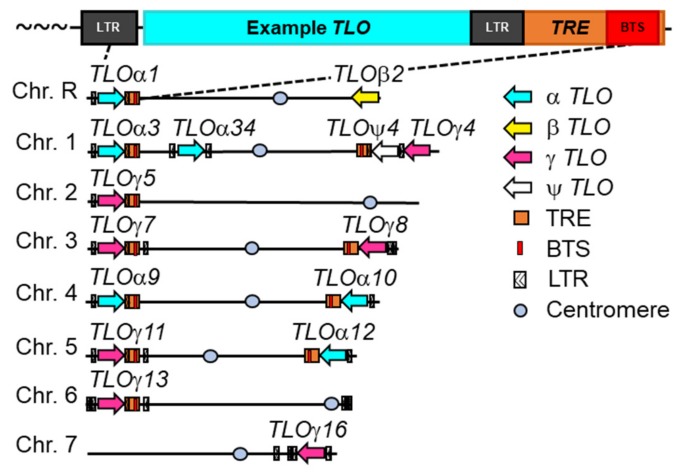
Organization of subtelomeric repetitive sequences in *C. albicans*. The telomere-associated (*TLO*) genes and other subtelomeric repetitive sequences are marked on each *C. albicans* chromosome arm for the genome reference strain SC5314. The *TLO* open reading frame (ORF; cyan) is commonly flanked by two long terminal repeats (LTR) elements, indicated in black. The *TLO* recombination element (TRE; orange) overlaps with the *TLO* 3’ untranslated region (UTR) and extends towards the centromere to encompass the Bermuda Triangle Sequence (BTS; red). All repetitive sequences are oriented similarly on all chromosome arms. Both the TRE and BTS contribute to subtelomeric recombination. Telomeric repeats are denoted by “~~~”. Figure adapted from [66].

**Figure 3 genes-10-00866-f003:**
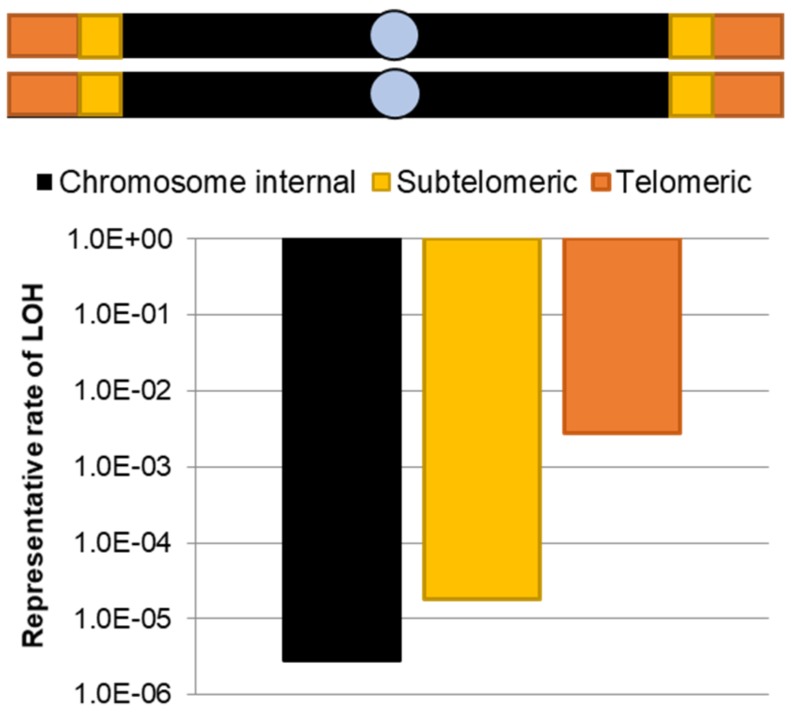
Rates of loss of heterozygosity (LOH) increase towards chromosome ends. An average LOH rate for each genomic region is depicted for chromosome internal sequences (black), the subtelomeres (yellow), and telomeres (orange) based on published (chromosome internal and subtelomeric) [28,56] and unpublished results (telomeric). All data is derived from LOH assays in which a *URA3* marker was inserted within different genomic regions and its location determined by either sequencing or contour clamped homogenous electric field (CHEF) gel analysis and Southern blotting. Centromeres are indicated by grey circles.

**Figure 4 genes-10-00866-f004:**
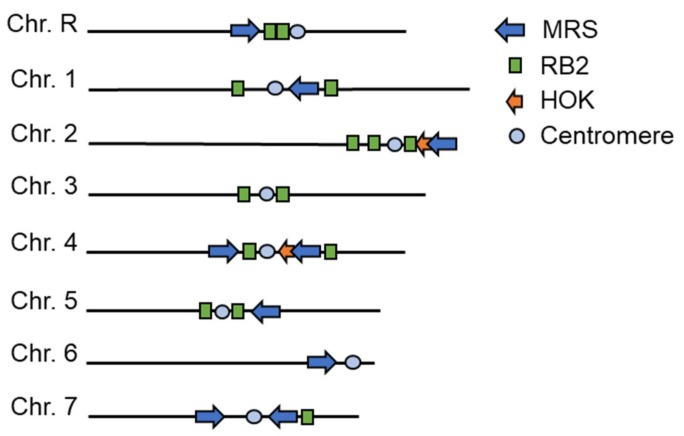
MRS elements in the *C. albicans* genome. The nine complete MRS loci (blue), 14 RB2 elements (green) and two HOK elements (orange) have been placed on their relative chromosome positions in *C. albicans* strain SC5314.

**Table 1 genes-10-00866-t001:** Repeat composition summary in *C. albicans*.

Repeat Element	Repeat Composition	Size	Repeat Copy Number (Haploid Genome)	Mechanisms of Instability
**Telomeres**	5’-ACTTCTTGGTGTACGGATGTCTA-3’	500 bp–5 kb	>20 repeats	Recombination, t-circles
**Subtelomeres**	Expanded gene families, LTRs, non-LTR retrotransposons	15 kb proximal to the telomeric repeats	14 *TLO* genes, 13 LTRs, 1 non-LTR retrotransposon	Recombination, LTR insertions, LOH, copy number variations
**Centromeres**	Unique regions without common sequence motifs, inverted repeats	Core 3 kb regions	Eight loci	Interchromosomal recombination, isochromosome formation
**Major Repeat Sequence**	RPS repeat units often flanked by non-repetitive RB2 and HOK elements	50 kb on average	Nine complete MRS loci, 14 RB2 elements, and two HOK elements	Chromosome translocations, intrachromosomal recombination
**rDNA locus**	18 s, 5.8s, 25 s, and 5 s rRNAs organized as tandem repeating units	11.6 kb - 12.5 kb per *RDN1* locus	21 to 176 copies of repeating *RDN1* locus	Intrachromosomal recombination
**ALS adhesin family**	Tandemly repeated Ser/Thr-rich domain	~3–6 kb	Eight genes	Intragenic recombination, intergenic recombination
**Long repeat sequences**	20 bp or longer repeats present more than once in the genome	65–6499 bp	1974 long repeats (2.87% of the haploid reference genome)	Recombination, LOH, chromosomal inversions, copy number variations

**Table 2 genes-10-00866-t002:** *C. albicans* characterized subtelomeric genes.

Gene Name	Description ^1^	Genomic Location
*TLO* **α** *1*	Member of a family of telomere-proximal genes of unknown function; hypha-induced expression; rat catheter biofilm repressed	Ca22chrRA_C_albicans_SC5314:9111 to 9863
**INO1**	Inositol-1-phosphate synthase; antigenic in human; repressed by farnesol in biofilm or by caspofungin; upstream inositol/choline regulatory element; glycosylation predicted; rat catheter, flow model induced; Spider biofilm repressed	Ca22chrRA_C_albicans_SC5314:2157044 to 2155482
*TLO* **β** *2*	Putative transcription factor/activator; Med2 mediator complex domain; transcript is upregulated in an RHE model of oral candidiasis; member of a family of telomere-proximal genes; Efg1, Hap43-repressed	Ca22chrRA_C_albicans_SC5314:2286198 to 2285377
**ATP16**	Subunit of the mitochondrial F1F0 ATP synthase; sumoylation target; protein newly produced during adaptation to the serum; Spider biofilm repressed	Ca22chrRA_C_albicans_SC5314:2285228 to 2284743
**XYL2**	D-xylulose reductase; immunogenic in mice; soluble protein in hyphae; induced by caspofungin, fluconazole, Hog1 and during cell wall regeneration; Mnl1-induced in weak acid stress; stationary phase enriched; flow model biofilm induced	Ca22chrRA_C_albicans_SC5314:2284454 to 2283372
**MAL2**	Alpha-glucosidase; hydrolyzes sucrose for sucrose utilization; transcript regulated by Suc1, induced by maltose, repressed by glucose; Tn mutation affects filamentous growth; upregulated in RHE model; rat catheter and Spider biofilm induced	Ca22chrRA_C_albicans_SC5314:2276745 to 2278457
**IAH1**	Protein similar to S. cerevisiae Iah1p, which is involved in acetate metabolism; mutation confers hypersensitivity to tunicamycin; transposon mutation affects filamentous growth	Ca22chrRA_C_albicans_SC5314:2276572 to 2275766
**CSM1**	Putative component of the monopolin complex with role in rDNA silencing, homologous chromosome segregation, protein localization to nucleolar rDNA repeats	Ca22chrRA_C_albicans_SC5314:2272097 to 2272774
*TLO* **α** *3*	Putative transcription factor; Med2 mediator domain; activates transcription in 1-hybrid assay in S. cerevisiae; repressed by Efg1; member of a family of telomere-proximal genes; Tbf1-induced	Ca22chr1A_C_albicans_SC5314:10718 to 11485
**TUP1**	Transcriptional corepressor; represses filamentous growth; regulates switching; role in germ tube induction, farnesol response; in repression pathways with Nrg1, Rfg1; farnesol upregulated in biofilm; rat catheter, Spider biofilm repressed	Ca22chr1A_C_albicans_SC5314:12163 to 13701
**MVD**	Mevalonate diphosphate decarboxylase; functional homolog of S. cerevisiae Erg19; possible drug target; regulated by carbon source, yeast-hypha switch, growth phase, antifungals; gene has intron; rat catheter, Spider biofilm repressed	Ca22chr1A_C_albicans_SC5314:13778 to 14917
*TLO* **γ** *4*	Member of a family of telomere-proximal genes of unknown function; transcript induced in an RHE model of oral candidiasis; Hap43-repressed	Ca22chr1A_C_albicans_SC5314:3186663 to 3186463
**FGR24**	Protein encoded in retrotransposon Zorro2 with similarity to retroviral endonuclease-reverse transcriptase proteins; lacks an ortholog in S. cerevisiae; transposon mutation affects filamentous growth	Ca22chr1A_C_albicans_SC5314:3185097 to 3184381
**SAM35**	Predicted component of the sorting and assembly machinery (SAM complex) of the mitochondrial outer membrane, involved in protein import into mitochondria	Ca22chr1A_C_albicans_SC5314:3177333 to 3176578
**GDI1**	Putative Rab GDP-dissociation inhibitor; GlcNAc-induced protein; Spider biofilm repressed	Ca22chr1A_C_albicans_SC5314:3172988 to 3171639
*TLO* **γ** *5*	Member of a family of telomere-proximal genes of unknown function; may be spliced in vivo	Ca22chr2A_C_albicans_SC5314:4248 to 4778
**RRN3**	Protein with a predicted role in recruitment of RNA polymerase I to rDNA; caspofungin induced; flucytosine repressed; repressed in core stress response; repressed by prostaglandins	Ca22chr2A_C_albicans_SC5314:5665 to 7335
**MPP10**	Putative SSU processome and 90S preribosome component; repressed in core stress response; repressed by prostaglandins	Ca22chr2A_C_albicans_SC5314:11063 to 9360
**FAV3**	Putative alpha-1,6-mannanase; induced by mating factor in MTLa/MTLa opaque cells	Ca22chr2A_C_albicans_SC5314:12443 to 11100
**PGA52**	GPI-anchored cell surface protein of unknown function; Hap43p-repressed gene; fluconazole-induced; possibly an essential gene, disruptants not obtained by UAU1 method	Ca22chr2A_C_albicans_SC5314:14415 to 13261
**RDH54**	Putative DNA-dependent ATPase with a predicted role in DNA recombination and repair; transcriptionally induced by interaction with macrophages	Ca22chr2A_C_albicans_SC5314:2221521 to 2223911
*TLO* **γ** *7*	Member of a family of telomere-proximal genes of unknown function; may be spliced in vivo; rat catheter biofilm repressed	Ca22chr3A_C_albicans_SC5314:13756 to 14265
*TLO* **γ** *8*	Member of a family of telomere-proximal genes of unknown function; may be spliced in vivo	Ca22chr3A_C_albicans_SC5314:1788114 to 1787605
**SRB1**	Essential GDP-mannose pyrophosphorylase; makes GDP-mannose for protein glycosylation; functional in S. cerevisiae psa1; on yeast-form, not hyphal cell surface; alkaline induced; induced on adherence to polystyrene; Spider biofilm repressed	Ca22chr3A_C_albicans_SC5314:1786177 to 1785089
*TLO* **α** *9*	Member of a family of telomere-proximal genes of unknown function; Hap43p-repressed gene	Ca22chr4A_C_albicans_SC5314:983 to 1660
**CNH1**	Na+/H+ antiporter; required for wild-type growth, cell morphology, and virulence in a mouse model of systemic infection; not transcriptionally regulated by NaCl; fungal-specific (no human or murine homolog)	Ca22chr4A_C_albicans_SC5314:5033 to 7435
**PHR3**	Putative beta-1,3-glucanosyltransferase with similarity to the A. fumigatus GEL family; fungal-specific (no human or murine homolog); possibly an essential gene, disruptants not obtained by UAU1 method	Ca22chr4A_C_albicans_SC5314:12793 to 14309
*TLO* **α** *10*	Member of a family of telomere-proximal genes of unknown function	Ca22chr4A_C_albicans_SC5314:1597812 to 1597159
**VID21**	Subunit of the NuA4 histone acetyltransferase complex; soluble protein in hyphae; Spider biofilm repressed	Ca22chr4A_C_albicans_SC5314:1596377 to 1594317
*TLO* **γ** *11*	Member of a family of telomere-proximal genes of unknown function; may be spliced in vivo	Ca22chr5A_C_albicans_SC5314:1918 to 2427
**CDC11**	Septin; cell and hyphal morphology, agar-invasive growth, full virulence and kidney tissue invasion in mouse, but not kidney colonization, immunogenicity; hyphal and cell-cycle-regulated phosphorylation; rat catheter biofilm repressed	Ca22chr5A_C_albicans_SC5314:8946 to 10154
**THS1**	Putative threonyl-tRNA synthetase; transcript regulated by Mig1 and Tup1; repressed upon phagocytosis by murine macrophages; stationary phase enriched protein; Spider biofilm repressed	Ca22chr5A_C_albicans_SC5314:14674 to 12554
*TLO* **α** *12*	Putative transcription factor; positive regulator of gene expression; Efg1-repressed; member of a family of telomere-proximal genes; transcript upregulated in RHE model of oral candidiasis	Ca22chr5A_C_albicans_SC5314:1182868 to 1182110
**HTS1**	Putative tRNA-His synthetase; downregulated upon phagocytosis by murine macrophage; stationary phase enriched protein; Spider biofilm repressed	Ca22chr5A_C_albicans_SC5314:1180902 to 1179397
**DES1**	Putative delta-4 sphingolipid desaturase; planktonic growth-induced gene	Ca22chr5A_C_albicans_SC5314:1178288 to 1179400
**EST1**	Telomerase subunit; allosteric activator of catalytic activity, but not required for catalytic activity; has TPR domain	Ca22chr5A_C_albicans_SC5314:1176356 to 1178194
**NRG2**	Transcription factor; transposon mutation affects filamentous growth	Ca22chr6A_C_albicans_SC5314:5 to 346
*TLO* **γ** *13*	Member of a family of telomere-proximal genes of unknown function; may be spliced in vivo; overlaps orf19.6337.1, which is a region annotated as blocked reading frame	Ca22chr6A_C_albicans_SC5314:5545 to 6069
**PGA25**	Putative GPI-anchored adhesin-like protein; fluconazole-downregulated; induced in oralpharyngeal candidasis; Spider biofilm induced	Ca22chr6A_C_albicans_SC5314:9894 to 7276
**HET1**	Putative sphingolipid transfer protein; involved in localization of glucosylceramide which is important for virulence; Spider biofilm repressed	Ca22chr6A_C_albicans_SC5314:12543 to 11950
**NAG4**	Putative transporter; fungal-specific; similar to Nag3p and to S. cerevisiae Ypr156Cp and Ygr138Cp; required for wild-type mouse virulence and wild-type cycloheximide resistance; gene cluster encodes enzymes of GlcNAc catabolism	Ca22chr6A_C_albicans_SC5314:1026875 to 1025130
**NAG3**	Putative MFS transporter; similar to Nag4; required for wild-type mouse virulence and cycloheximide resistance; in gene cluster that includes genes encoding enzymes of GlcNAc catabolism; Spider biofilm repressed	Ca22chr6A_C_albicans_SC5314:1024922 to 1023237
**DAC1**	N-acetylglucosamine-6-phosphate (GlcNAcP) deacetylase; N-acetylglucosamine utilization; required for wild-type hyphal growth and virulence in mouse systemic infection; gene and protein are GlcNAc-induced; Spider biofilm induced	Ca22chr6A_C_albicans_SC5314:1021987 to 1023228
**NAG1**	Glucosamine-6-phosphate deaminase; required for normal hyphal growth and mouse virulence; converts glucosamine 6-P to fructose 6-P; reversible reaction in vitro; gene and protein is GlcNAc-induced; Spider biofilm induced	Ca22chr6A_C_albicans_SC5314:1021746 to 1021000
**HXK1**	N-acetylglucosamine (GlcNAc) kinase; involved in GlcNAc utilization; required for wild-type hyphal growth and mouse virulence; GlcNAc-induced transcript; induced by alpha pheromone in SpiderM medium	Ca22chr6A_C_albicans_SC5314:1019276 to 1020757
**RVS161**	Protein required for endocytosis; contains a BAR domain, which is found in proteins involved in membrane curvature; null mutant exhibits defects in hyphal growth, virulence, cell wall integrity, and actin patch localization	Ca22chr7A_C_albicans_SC5314:2040 to 1246
**RAD3**	Ortholog of S. cerevisiae Rad3; 5′ to 3′ DNA helicase, nucleotide excision repair and transcription, subunit of RNA polII initiation factor TFIIH and Nucleotide Excision Repair Factor 3 (NEF3)	Ca22chr7A_C_albicans_SC5314:5761 to 3464
**SAC7**	Putative GTPase activating protein (GAP) for Rho1; repressed upon adherence to polystyrene; macrophage/pseudohyphal-repressed; transcript is upregulated in RHE model of oral candidiasis and in clinical oral candidiasis	Ca22chr7A_C_albicans_SC5314:9430 to 7583
**CSA1**	Surface antigen on elongating hyphae and buds; strain variation in repeat number; ciclopirox, filament induced, alkaline induced by Rim101; Efg1-, Cph1, Hap43-regulated; required for WT RPMI biofilm formation; Bcr1-induced in a/a biofilms	Ca22chr7A_C_albicans_SC5314:13080 to 10024
**FRP2**	Putative ferric reductase; alkaline induced by Rim101; fluconazole-downregulated; upregulated in the presence of human neutrophils; possibly adherence-induced; regulated by Sef1, Sfu1, and Hap43	Ca22chr7A_C_albicans_SC5314:14047 to 15825
*TLO* **γ** *16*	Member of a family of telomere-proximal genes of unknown function; may be spliced in vivo	Ca22chr7A_C_albicans_SC5314:943352 to 942825
**SYS1**	Putative Golgi integral membrane protein; transcript regulated by Mig1	Ca22chr7A_C_albicans_SC5314:941499 to 940993
**SPT6**	Putative transcription elongation factor; transposon mutation affects filamentous growth; transcript induced in an RHE model of oral candidiasis and in clinical isolates from oral candidiasis	Ca22chr7A_C_albicans_SC5314:934878 to 939083

^1^ As assigned in the Candida Genome Database [49].

**Table 3 genes-10-00866-t003:** *C. albicans* subtelomeric LTRs and retrotransposons.

ORF Name	Description ^1^	Genomic Location
**gamma-1c**	Solo copy of the long terminal repeat (LTR) associated with the transposon Tca2; about 280 bp long, 5-10 copies per genome	Ca22chr1A_C_albicans_SC5314:3187028 to 3187306
**gamma-1b**	Solo copy of the long terminal repeat (LTR) associated with the transposon Tca2; about 280 bp long, 5-10 copies per genome	Ca22chr1A_C_albicans_SC5314:3187383 to 3187662
**Zorro2-1**	Non-LTR retrotransposon, encodes a potential DNA-binding zinc-finger protein and a polyprotein similar to pol with conserved endonuclease and reverse transcriptase domains; member of L1 clade of transposons	Ca22chr1A_C_albicans_SC5314:3185887 to 3181663
**san-1a**	Solo copy of the long terminal repeat (LTR) associated with the transposon Tca4; about 381 bp long, 1-4 copies per genome	Ca22chr1A_C_albicans_SC5314:3178575 to 3178194
**rho-2a**	Long terminal repeat (LTR); about 275 bp long, 13 copies per genome	Ca22chr2A_C_albicans_SC5314:4611 to 4885
**rho-3a**	Long terminal repeat (LTR); about 275 bp long, 13 copies per genome	Ca22chr3A_C_albicans_SC5314:14098 to 14372
**tui-3a**	Long terminal repeat (LTR); about 199 bp long, 19 copies per genome	Ca22chr3A_C_albicans_SC5314:14751 to 14553
**rho-3b**	Long terminal repeat (LTR); about 275 bp long, 13 copies per genome	Ca22chr3A_C_albicans_SC5314:1787772 to 1787498
**rho-5a**	Long terminal repeat (LTR); about 275 bp long, 13 copies per genome	Ca22chr5A_C_albicans_SC5314:2260 to 2534
**rho-6a**	Long terminal repeat (LTR); about 275 bp long, 13 copies per genome	Ca22chr6A_C_albicans_SC5314:5902 to 6176
**chi-6a**	Long terminal repeat (LTR) associated with the transposon Tca10; about 192 bp long, 11 copies per genome	Ca22chr6A_C_albicans_SC5314:1032389 to 1032200
**kappa-6a**	Long terminal repeat (LTR) associated with the transposon Tca6; about 280 bp long, 10-15 copies per genome	Ca22chr6A_C_albicans_SC5314:1028505 to 1028227
**weka-6a**	Long terminal repeat (LTR); about 167 bp long, 9 copies per genome	Ca22chr6A_C_albicans_SC5314:1027905 to 1028066
**san-7a**	Solo copy of the long terminal repeat (LTR) associated with the transposon Tca4; about 381 bp long, 1-4 copies per genome	Ca22chr7A_C_albicans_SC5314:942855 to 942475

^1^ As assigned in the Candida Genome Database [49].

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
