# Peer review of "To Repeat or Not to Repeat: Repetitive Sequences Regulate Genome Stability in Candida albicans"

_genes, 2019, doi:10.3390/genes10110866_

Round 1

Reviewer 1 Report

Overall, this is a comprehensive and informative review about repeat structures in the Candida albicans genome with an interesting discussion about how these repeats contribute to genome instability and pathogen evolution. It should be of interest to a variety of readers in fungal biology and genome biology fields. However, I have a few suggestions for improving clarity and understandability for the reader. My specific comments are as follows:

A reference for fern chromosome number should be included in the first paragraph.

Define rDNA first usage (in abstract) and give abbreviation (first use in text).

The use of past tense in the budding yeast paragraph is confusing since the review focuses on recent data for one of the budding yeasts. Additionally, given the major contributions to repeats of non-budding fungi, I think it would make more sense to introduce this idea as fungi more broadly. Furthermore, a sentence or two more about how fungi have contributed to our understanding would help to make this transition clearer.

The paragraphs on telomere maintenance (2.1) read a bit like a list of separate facts about proteins involved in C. albicans telomeres. Think about how to describe the structures more holistically. Perhaps combinining 2.1 and 2.2 together into a discussion of what the structures are and how they prevent or promote replication or recombination might make the issue easier to understand for the author. A figure could be very useful for helping to visualize these processes (instead of the current figure 2 – see below).

The supplementary table 1 with all of the sub-telomeric genes and some annotation information is a very useful reference. Similarly, the summary of X number of 55 genes with Y function should be kept in the text. However, the word cloud figure for figure 2 seems pointless to me as it doesn’t add any additional information.

How much are retrotransposons enriched in sub-telomeres? If 350 total insertions and 14 are in the sub-telomeres, how much is this higher than would be expected on a per kb basis? My quick calculation says about 2X higher than would be expected, but this would be good information to include.

The discussion seems to end abruptly. A bit more synthesis about the implications of the different types of repeats and mechanisms would help the review seem more cohesive to the reader.

Author Response

Reviewer 1:

A reference for fern chromosome number should be included in the first paragraph.

This has been included per the Reviewer’s suggestion (line 28).

Define rDNA first usage (in abstract) and give abbreviation (first use in text).

This has been edited per the Reviewer’s suggestion (lines 14-15).

The use of past tense in the budding yeast paragraph is confusing since the review focuses on recent data for one of the budding yeasts. Additionally, given the major contributions to repeats of non-budding fungi, I think it would make more sense to introduce this idea as fungi more broadly. Furthermore, a sentence or two more about how fungi have contributed to our understanding would help to make this transition clearer.

These suggestions have been incorporated this paragraph to provide more context by beginning with the importance of fungi to eukaryotic biology, then to analysis of repetitive sequences, and then comparative genomics (lines 56-61).

The paragraphs on telomere maintenance (2.1) read a bit like a list of separate facts about proteins involved in C. albicans telomeres. Think about how to describe the structures more holistically. Perhaps combining 2.1 and 2.2 together into a discussion of what the structures are and how they prevent or promote replication or recombination might make the issue easier to understand for the author. A figure could be very useful for helping to visualize these processes (instead of the current figure 2 – see below).

Reviewer 1 has aptly pointed out the incomplete picture of telomere regulation that exists in C. albicans. Although a number of proteins have been identified to operate in telomere length regulation, how they contribute to this function is unknown and many of the canonical complexes have not been resolved. Significant differences in function among telomere proteins and the complete absence of a number of factors (e.g., Sir proteins) makes translating events in other yeasts and fungi difficult. Per the suggestion of Reviewers 1 and 3, we have included information regarding the ALT telomere lengthening process as is observed in S. cerevisiae, although are unable to comment further on these processes in C. albicans due to the lack of further directed research into this process of telomere regulation.

The supplementary table 1 with all of the sub-telomeric genes and some annotation information is a very useful reference. Similarly, the summary of X number of 55 genes with Y function should be kept in the text. However, the word cloud figure for figure 2 seems pointless to me as it doesn’t add any additional information.

We agree with Reviewer 1 (and Reviewer 2) and have removed Figure 2 from the manuscript, leaving the in-text summary.

How much are retrotransposons enriched in sub-telomeres? If 350 total insertions and 14 are in the sub-telomeres, how much is this higher than would be expected on a per kb basis? My quick calculation says about 2X higher than would be expected, but this would be good information to include.

We thank both Reviewer 1 and Reviewer 3 for their suggestion to include the increased rate of LTR insertion in the subtelomeres and have added it into the text (lines 211-214).

The discussion seems to end abruptly. A bit more synthesis about the implications of the different types of repeats and mechanisms would help the review seem more cohesive to the reader.

We agree that the Conclusions can be expanded to synthesize larger ideas regarding repeat structure or location and production of novel genotypes. We have included this as part of the new Discussion (lines 391-404).

Reviewer 2 Report

This review article by “Dunn and Anderson” describes the four major types of repetitive sequences i.e. telomeres, subtelomeres, the Major Repeat Sequences, and the rDNA locus in the genome of candida albicans (an opportunistic human fungal pathogen) and highlights their contribution in the mutational processes of genome stability and evolution. Furthermore, authors include information about additional repetitive sequences in candida albicans genome that play an important role in the promotion of recombination.

The article is a fun read and present interesting literature on the repeat sequences in candida albicans. The writing is engaging and overall the manuscript is structured very nicely.

Comments are as follows:

Figures : I have several comments that authors may consider:

Figure 2 (Gene descriptos for C.albicans subtelomeric genes) : It's hard to distinguish the grey from black (especially in the printed copy of the paper).

Similarily, in figure 3 green appears to be the cyan colour (to this reviewer) and please cite references 60 and 45 ( Or should include a statement “figure adapted from references 60 and 45).

Line 47-50, reference 6 is irrelevant and should be deleted.

Line 102, “telomerase associated proteins, Est1 and Est3”. Did authors mean “protein subunits, named Est1p and Est3p” as described in cited reference 34?

Line 207, please describe the term “tui-class” for the inexperienced readers.

Line 257-258, It would be interesting to include information about whether this truncated TLO gene expressed or not.

Line 321. Provide the full form of ALS in the subtitle “The ALS family of adhesins”.

Author Response

Reviewer 2:

Figure 2 (Gene descriptors for C. albicans subtelomeric genes) : It's hard to distinguish the grey from black (especially in the printed copy of the paper).

We have elected to remove Figure 2 from the manuscript per suggestions from Reviewer 1.

Similarly, in figure 3 green appears to be the cyan colour (to this reviewer) and please cite references 60 and 45 (Or should include a statement “figure adapted from references 60 and 45).

We have corrected the figure legend to indicate the appropriate color in Figure 3. A statement has also been included to indicate adaptation of this figure from Moran et al, Curr Genet, 2019.

Line 47-50, reference 6 is irrelevant and should be deleted.

This has been removed per the Reviewer’s suggestion.

Line 102, “telomerase associated proteins, Est1 and Est3”. Did authors mean “protein subunits, named Est1p and Est3p” as described in cited reference 34?

This has been edited per the Reviewer’s suggestion.

Line 207, please describe the term “tui-class” for the inexperienced readers.

Two steps have been taken to clarify the tui-class LTR referenced here. First, we reorganized the section in lines 207-209 to provide more detail for the naming of different LTR classes based on sequence. Second, we connected this point to the sentence on line 229 by linking these sequences as one LTR family of the 34 present in the C. albicans genome.

Line 257-258, It would be interesting to include information about whether this truncated TLO gene expressed or not.

This information has been included per the Reviewer’s request (line 282) in which we note TLOg4 is actively transcribed while the TLOy4 pseudogene is silent.

Line 321. Provide the full form of ALS in the subtitle “The ALS family of adhesins”.

This has been edited per the Reviewer’s suggestion (line 346).

Reviewer 3 Report

The review by Dunn and Anderson summarizes various aspects of contemporary knowledge on the repetitive genome regions in the genome of opportunistic pathogenic yeast Candida albicans. Overall contents, structure and style of writing do not raise any issues.

One minor gap of the review is the lack of systematic overview of repeat composition of the reference genome, like names and copy numbers for major classes of transposons and tandem repeats. Some of these data are scattered across various sections.

section 3.3 - it is unclear how to infer LOH at telomeres, which consist of tandemly repeated sequences (which are identical not only between homologs, but also across all chromosomes). The articles cited only estimate LOH rate for subtelomeres and internal chromosome regions, so I assume the estimate for telomeres belongs to "unpublished results" referenced in Fig.4 caption. Please clarify.

section 3.2 - enrichment of repetitive elements in subtelomeres can be quantified based on Table 2 data: 14/~350 insertions are located in 16*0.015=0.24Mbp of 15.47Mbp genome assembly - ca. 2.6 fold enrichment.

Minor points:
line 86 - please de-abbreviate genus name for D. hansenii
line 96 - "DNA replication" does not seem an appropriate term for telomerase activity
line 114-115 - the role of recombination in the genome instability during telomere shortening is unclear to me
line 199 - italicize species name
section 4 - in text RB2 and HOK are included in MRS structure, while on figure those are separate.
lines 319-320 "additional sequence elements ... promote recombination" - consider rewording

Author Response

Reviewer 3:

One minor gap of the review is the lack of systematic overview of repeat composition of the reference genome, like names and copy numbers for major classes of transposons and tandem repeats. Some of these data are scattered across various sections.

Per Reviewer 3’s suggestion we have included a summary table, Table 1, to highlight the major repetitive sequences/regions in C. albicans covered in this review.

section 3.3 - it is unclear how to infer LOH at telomeres, which consist of tandemly repeated sequences (which are identical not only between homologs, but also across all chromosomes). The articles cited only estimate LOH rate for subtelomeres and internal chromosome regions, so I assume the estimate for telomeres belongs to "unpublished results" referenced in Fig.4 caption. Please clarify.

The Figure 3 legend has been clarified per the Reviewer’s request to describe specifically that the telomeric LOH rate is from unpublished data. Additional details describing the general methodology applied to determine these rates has also been added for clarity.

section 3.2 - enrichment of repetitive elements in subtelomeres can be quantified based on Table 2 data: 14/~350 insertions are located in 16*0.015=0.24Mbp of 15.47Mbp genome assembly - ca. 2.6 fold enrichment.

We thank both Reviewer 1 and Reviewer 3 for their suggestion to include the increased rate of LTR insertion in the subtelomeres (line 211-212).

line 86 - please de-abbreviate genus name for D. hansenii

This has been edited per the Reviewer’s suggestion (line 95).

line 96 - "DNA replication" does not seem an appropriate term for telomerase activity

We agree with the Reviewer and have edited this to say “reverse transcription” (line 104).

line 114-115 - the role of recombination in the genome instability during telomere shortening is unclear to me

Per the suggestion of Reviewers 1 and 3, we have included information regarding recombination during telomere shortening and, specifically, hints at alternative lengthening of telomere (ALT) operating in telomere length regulation in C. albicans. We acknowledge within this section (2.2) that major holes exist in current models of regulation due to a lack of research into these topics.

line 199 - italicize species name

This has been edited per the Reviewer’s suggestion.

section 4 - in text RB2 and HOK are included in MRS structure, while on figure those are separate.

We have edited this sentence in the text to clarify these elements are technically independent but commonly associated with the MRS (line 286).

lines 319-320 "additional sequence elements ... promote recombination" - consider rewording

We have restructured this sentence for clarity (lines 343-344).